# Multi-Scale Patch Transformer Network for Satellite Image Time Series Classification

Jifeng Suo
*School of Control Science and Engineering*
*Dalian University of Technology*
Dalian, China
suojifeng@mail.dlut.edu.cn

Degang Wang
*School of Control Science and Engineering*
*Dalian University of Technology*
Dalian, China
wangdg@dlut.edu.cn

*Abstract*—**With the increasing availability of high-quality earth observation data, satellite image time series (SITS) classification has become a hot topic. In this paper, a multi-scale patch Transformer model (PatchSITS) for SITS classification is proposed. First, SITS samples are segmented into patch sequences of varying patch lengths using k-means clustering. Subsequently, the enhanced Transformer is proposed to capture temporal features at various scales. To capture inter-band relationships and enhance critical band information, the gated channel attention mechanism is applied to obtain dynamic weights between bands. Furthermore, a multi-scale weighted fusion strategy is proposed to integrate these multi-scale features. And broad learning system (BLS) is utilized for SITS classification. Numerical simulations demonstrate that the PatchSITS model exhibits excellent classification performance on the BreizhCrops dataset.**

*Keywords—multi-scale temporal patching, crop classification, satellite image time series, transformer, k-means cluster*

## I. INTRODUCTION

With continuous advancements in remote sensing technology and the growing number of satellites, high-quality earth observation data can be obtained conveniently, which provides a vast development space for SITS classification. The SITS is utilized to analyze crop growth cycles and environmental changes, which can help agricultural management and improve the accuracy of crop classification.

To address the crop classification problem in SITS, numerous methods have been proposed. Due to the advantages in extracting and classifying features, various deep learning models are widely applied in SITS data analysis ([1]). Recurrent neural networks (RNNs) are specifically designed for handling temporal data and capturing contextual information in SITS. Bidirectional long short-term memory networks (LSTM) are applied to identify temporal patterns and trends in SITS classification ([2]). To address the common issues of gradient vanishing and explosion in deep RNN architectures for crop classification, a novel stackable RNN unit called StarRNN is proposed to capture long-term dependencies ([3]). In contrast to recurrent neural networks that employ an encoder only, an encoder-decoder structure based on RNN sequences is designed to approximate the growth model of vegetation classes ([4]). Given the good performance of convolutional neural networks (CNNs) in capturing spatial features of remote sensing data, TempCNN is designed to extract temporal and spectral features from SITS data through convolution operations across the temporal and spectral dimensions ([5]). In addition, a temporal convolutional network (CA-TCN) based on channel attention and temporal convolution module is proposed for crop classification ([6]). A multi-scale temporal Transformer convolution (Ms-TTC) network model is designed to integrate multi-scale global and local information for effective processing of complex SITS data ([7]). To further address the challenge of single-level category prediction, a convolutional LSTM model is implemented with the goal of predicting hierarchical labels for each pixel ([8]).

Due to the strong capability of attention-based networks in capturing global information, a Transformer network is specifically developed to model long-term dependencies and global features for crop classification ([9]). In contrast to conventional Transformers, the temporal attention encoder model combines pixel-set encoders with temporal self-attention mechanisms to effectively capture both global and local features in SITS data. To improve the extraction of local semantic information, the patch Transformer (PatchTST) model segments multivariate time series data into smaller patches for more effective modeling ([11]). Based on this framework, the multi-resolution time series Transformer model incorporates a Transformer architecture with relative position encoding, thereby enhancing its ability to process multivariate time series data across various temporal resolutions ([12]).

Now, remote sensing data can be available for a period of time, hence how to effectively utilize multi-scale temporal information for improving classification performance is an interesting question. In this paper, a multi-scale patch Transformer model is designed to extract the feature of SITS and BLS ([13]) is considered for the classifier. The main contributions are that:

- A multi-scale Patch Transformer feature extraction model, with scales determined using k-means clustering, is proposed to effectively capture the temporal information in satellite image time series.

- A multi-scale weighted feature fusion classification model is designed to effectively utilize multi-scale temporal remote sensing information and improve the accuracy of the classification process.

This paper is organized as follows. In Section II, the feature extraction method for multi-scale patch sequences in SITS is

designed. In Section III, a multi-scale weighted feature fusion method is designed and BLS is considered as a classifier for crop remote sensing data. Numerical experiments are presented in Section IV. Conclusions are summarized in Section V.

## II. MULTI-SCALE PATCH SEQUENCE FEATURE EXTRACTION IN SATELLITE IMAGE TIME SERIES

In Section II, a feature extraction method for multi-scale patch sequences based on the enhanced Transformer model is designed. The specific architecture of the PatchSITS model is illustrated in Fig. 1. The model primarily consists of the following components: the multi-scale patch embedding module, the enhanced Transformer module, and the multi-scale feature fusion module.

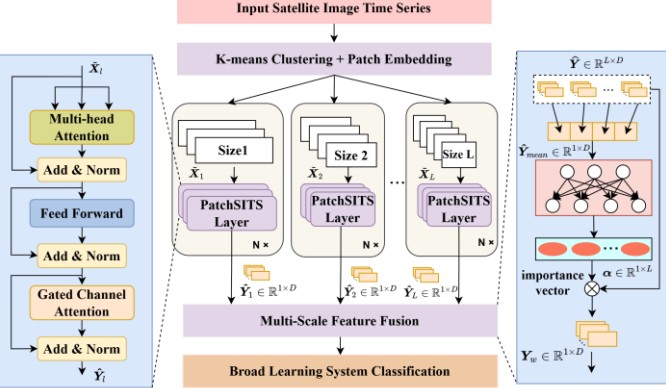

Fig. 1. Architecture of the PatchSITS model.

In the patch embedding module, the input satellite image time series is divided into multi-scale sub-sequence patches, where the length of the patch is computed by the k-means clustering algorithm ([14]). The enhanced Transformer module is used for multi-scale feature extraction, effectively capturing both local and global temporal information. A gated channel attention is designed to dynamically obtain the dynamic weights between bands.

### A. K-means Clustering and Patch Embedding

In the task of crop classification, the types of crops can be identified through the segmentation of SITS which captures the local changes in the crop growth cycle. Since single-scale patch segmentation cannot fully capture crop growth information, the multi-scale patch segmentation strategy is proposed.

The set of multi-scale patch lengths is determined by applying a k-means clustering algorithm to features with different patch lengths. The input SITS samples $X = [x_1, x_2, ..., x_T]^T \in \mathbb{R}^{T \times C}$ consist of $T$ time steps, where $x_t \in \mathbb{R}^{1 \times C}$ represents the feature vector at time step $t (t = 1,2, ..., T)$, and $C$ represents the number of bands at each time step.

To determine the optimal patch length for segmentation, a set of pre-selected patch lengths is defined as $\{P_1, P_2 ..., P_G\}$, where $P_G = T/2$. The clustering process for the patch length $P_g$ is taken as an example. The input samples $X$ are transformed into a sequence of patches $X_g \in \mathbb{R}^{C \times N_g \times P_g}$ using the sliding window method, where $N_g$ represents the number of patches and $P_g$ denotes the patch length. $N_g$ is calculated by the following equation:

$$N = \left\lfloor \frac{T - P}{S} \right\rfloor + 2 \tag{1}$$

For the patch sequence $X_g$, the corresponding sub-patch sequences are denoted by $X_{g,i}$. The relevant clustering features $\hat{X}_{g,i}$ are calculated, including the band mean $X_{g,i,mean}$, band variance $X_{g,i,var}$, Fourier coefficients $X_{g,i,fft}$, and normalized difference vegetation index $X_{g,i,ndvi}$. The band mean $X_{g,i,mean}$ represents the average value across different bands. The band variance $X_{g,i,var}$ refers to the variance across the bands. The calculation equations are as follows:

$$X_{g,i,mean} = 1/P_g \sum_{i=1}^{P_g} X_{g,i} \tag{2}$$

$$X_{g,i,var} = 1/P_g \sum_{i=1}^{P_g} (X_{g,i} - X_{g,i,mean})^2 \tag{3}$$

$$X_{g,i,fft} = \text{mean}(|FFT(X_{g,i})|) \tag{4}$$

$$X_{g,i,ndvi} = NIR_{g,i} - RED_{g,i}/NIR_{g,i} + RED_{g,i} \tag{5}$$

Thus, the clustering features $\hat{X}_{g,i}$ for the patch sub-sequences $X_{g,i}$ are represented by concatenating:

$$\hat{X}_{g,i} = (X_{g,i,mean}, X_{g,i,var}, X_{g,i,fft}, X_{g,i,ndvi}) \tag{6}$$

Similar patches are clustered using the k-means clustering algorithm based on the feature $\hat{X}_{g,i}$. The quality of clustering is evaluated using the Silhouette score $S_g$ for the patch length $P_g$, which is computed by averaging the Silhouette scores $S_{g,i}$ of all patches within each cluster, where $i = 1,2, ..., N_g$. The Silhouette score $S_{g,i}$ is calculated as follows:

$$S_{g,i} = \frac{a_{g,i} - b_{g,i}}{max(a_{g,i}, b_{g,i})} \tag{7}$$

where $a_{g,i}$ denotes the average feature distance between the $i$-th patch and all other patches within the same cluster, while $b_{g,i}$ represents the minimum average distance between the $i$-th patch and the patches in any other cluster.

$$\begin{cases} a_{g,i} = \frac{1}{|C_i| - 1} \sum_{j \in C_i, j \neq i} d(\hat{X}_{g,i}, \hat{X}_{g,j}) \\ b_{g,i} = \min_{k \neq C_i}(\frac{1}{|C_k|} \sum_{j \in C_k} d(\hat{X}_{g,i}, \hat{X}_{g,j})) \end{cases} \tag{8}$$

where $C_i$ denotes the cluster to which the $i$-th sample belongs, $C_k$ denotes clusters other than the one to which the $i$-th sample belongs, $\hat{X}_{g,i}$ and $\hat{X}_{g,j}$ are the feature vectors of the $i$-th and $j$-th samples, and $d(\hat{X}_{g,i}, \hat{X}_{g,j})$ denotes the distance between them. A higher Silhouette score means that the selected patch length can effectively capture the primary patterns in the data.

Based on the pre-selected patch length set $\{P_1, P_2 ..., P_G\}$, a set of Silhouette scores $\{S_1, S_2 ..., S_G\}$ can be obtained. The top $L$ patch lengths with the highest Silhouette scores are selected to form the multi-scale patch length set $P = \{P_1, P_2 ..., P_L\}$.

The steps for selecting patch lengths through clustering are summarized as follows:

- Step 1. Define the SITS sample set $\boldsymbol{\mathcal{X}} = \{\boldsymbol{X}_1, \boldsymbol{X}_2, \dots, \boldsymbol{X}_M\}$ and a set of pre-selected patch lengths $\{P_1, P_2 \dots, P_G\}$.

- Step 2. For each patch length in $\{P_1, P_2 \dots, P_G\}$, extract and standardize features from the samples in $\boldsymbol{\mathcal{X}}$.

- Step 3. Compute the Silhouette scores for each patch length in $\{P_1, P_2 \dots, P_G\}$.

- Step 4. Rank the patch lengths based on their Silhouette scores, and select the top $L$ patch lengths to form the multi-scale patch set $\boldsymbol{P} = \{P_1, P_2 \dots, P_L\}$.

Through the above steps, the multi-scale patch set $\boldsymbol{P} = \{P_1, P_2 \dots, P_L\}$ is selected. The input SITS data $\boldsymbol{X} \in \mathbb{R}^{T \times C}$ is divided into patch data of various scales $\{\widehat{\boldsymbol{X}}_1, \widehat{\boldsymbol{X}}_2 \dots, \widehat{\boldsymbol{X}}_L\}$, where $\widehat{\boldsymbol{X}}_l \in \mathbb{R}^{C \times N_l \times P_l}$ represents a sequence of $N_l$ patches. The value of $N_l$ is calculated by equation (1).

The single-channel strategy is used to extract the unique information from each band, ensuring that the specific characteristics of each band are fully captured. In this approach, the single-band satellite image time series $\widehat{\boldsymbol{X}}_{l,c} \in \mathbb{R}^{N_l \times P_l}$ is used as the input for the $c$-th band feature (where $c = 1,2, \dots, C$). $\widehat{\boldsymbol{X}}_{l,c}$ is input into the projection mapping layer to increase the flexibility of the PatchSITS model in processing data. This process maps the input $\widehat{\boldsymbol{X}}_{l,c}$ to the latent vector $\widetilde{\boldsymbol{X}}_{l,c} \in \mathbb{R}^{N_l \times D}$ by the following equation:

$$\widetilde{\boldsymbol{X}}_{l,c} = \widehat{\boldsymbol{X}}_{l,c}\boldsymbol{W}_l + \boldsymbol{W}_{pos} \tag{9}$$

where $\boldsymbol{W}_l \in \mathbb{R}^{P_l \times D}$ is the weight parameter, $\boldsymbol{W}_{pos} \in \mathbb{R}^{N_l \times D}$ is the position encoding, and $N_l$ is the number of patches after $l$-scale division. The position encoding $\boldsymbol{W}_{pos}$ simulates positional information by mapping each patch to a fixed vector with distinct frequencies ([15]) as follows:

$$\boldsymbol{W}_{pos} = \begin{cases} sin(pos/10000^{2k/D}) & if\ i = 2k \\ cos(pos/10000^{2k/D}) & if\ i = 2k+1 \end{cases} \tag{10}$$

where $pos$ is the position index in the current patch sequence, $i$ represents the feature dimension index, and $D$ is the dimension of the latent vector.

### B. Enhanced Transformer

The enhanced Transformer layer primarily comprises of the multi-head attention layer, the feed-forward network layer, and the gated channel attention layer.

The multi-head attention mechanism employs $h$ parallel attention heads to capture various aspects of SITS and enhance feature representation, using $\widetilde{\boldsymbol{X}}_l = (\widetilde{\boldsymbol{X}}_{l,1}, \widetilde{\boldsymbol{X}}_{l,2}, \dots \widetilde{\boldsymbol{X}}_{l,c})$ as the input. Similarity between patches in the output data sequence is determined by equations (11) and (12). Three independent linear transformations are applied to create the query matrix, the key matrix, and the value matrix, thereby capturing various features and patterns in the data. For the $h$-th attention head, the equations for the query matrix $\boldsymbol{Q}_{l,c,h} \in \mathbb{R}^{N_l \times d_k}$, the key matrix $\boldsymbol{K}_{l,c,h} \in \mathbb{R}^{N_l \times d_k}$, and the value matrix $\boldsymbol{V}_{l,c,h} \mathbb{R}^{N_l \times d_v}$ are as follows:

$$\boldsymbol{Q}_{l,c,h} = \widetilde{\boldsymbol{X}}_{l,c}\boldsymbol{W}_h^Q, \boldsymbol{K}_{l,c,h} = \widetilde{\boldsymbol{X}}_{l,c}\boldsymbol{W}_h^K, \boldsymbol{V}_{l,c,h} = \widetilde{\boldsymbol{X}}_{l,c}\boldsymbol{W}_h^V \tag{11}$$

where $\boldsymbol{W}_h^Q \in \mathbb{R}^{D \times d_k}$, $\boldsymbol{W}_h^K \in \mathbb{R}^{D \times d_k}$, $\boldsymbol{W}_h^V \in \mathbb{R}^{D \times d_v}$ are weight parameters, $d_v = d_k = D/h$. The attention output $\boldsymbol{O}_{l,c,h} \in \mathbb{R}^{N_l \times d_v}$ for the $h$-th attention head is computed by the attention function using the query matrix $\boldsymbol{Q}_{l,c,h}$ the key matrix $\boldsymbol{K}_{l,c,h}$, and the value matrix $\boldsymbol{V}_{l,c,h}$:

$$\boldsymbol{O}_{l,c,h} = \phi_{softmax}\left(\frac{\boldsymbol{Q}_{l,c,h}\boldsymbol{K}_{l,c,h}^{\mathrm{T}}}{\sqrt{d_k}}\right)\boldsymbol{V}_{l,c,h} \tag{12}$$

where $\phi_{softmax}(\cdot)$ is the softmax function.

To obtain the multi-head attention output, the outputs from single attention heads are combined along the dimension $h$ and subsequently passed through a linear transformation. This process enhances the learning capability of the model and enrich feature representation. The output of multi-head attention $\boldsymbol{O}_{l,c} \in \mathbb{R}^{N_l \times D}$ can be determined using the following equation:

$$\boldsymbol{O}_{l,c} = (\boldsymbol{O}_{l,c,1}, \boldsymbol{O}_{l,c,2}, \dots, \boldsymbol{O}_{l,c,h})\boldsymbol{W}^O \tag{13}$$

where the weight parameter of the linear transformation is $\boldsymbol{W}^O \in \mathbb{R}^{D \times D}$. The residual connection is used to connect the input data $\widetilde{\boldsymbol{X}}_{l,c}$ with the output data $\boldsymbol{O}_{l,c}$ for addressing the vanishing gradients and enhancing model performance. The combined result $\widetilde{\boldsymbol{O}}_{l,c} \in \mathbb{R}^{N_l \times D}$ is passed through a normalization layer to standardize the data:

$$\widetilde{\boldsymbol{O}}_{l,c} = \varphi(\boldsymbol{O}_{l,c} + \widetilde{\boldsymbol{X}}_{l,c}) \tag{14}$$

where $\varphi(\cdot)$ represents the LayerNorm normalization layer, which is used to adjust the feature distribution. Given the input vector $\boldsymbol{O}_{l,c} \in \mathbb{R}^{N_l \times D}$, the normalized result is calculated as follows:

$$\varphi(\boldsymbol{O}_{l,c}) = \gamma \frac{(\boldsymbol{O}_{l,c} - \mu_{\boldsymbol{O}_{l,c}})}{\sqrt{\sigma_{\boldsymbol{O}_{l,c}}^2 + \varepsilon}} + \beta \tag{15}$$

where $\mu_{\boldsymbol{O}_{l,c}}$ and $\sigma_{\boldsymbol{O}_{l,c}}^2$ represent the mean and variance of the input vector $\boldsymbol{O}_{l,c}$ across the $D$ dimension. The parameters $\gamma$ and $\beta$ are applied to scale and shift the normalized features.

Additionally, the normalized output data $\widetilde{\boldsymbol{O}}_{l,c}$ is input into a feedforward network layer to extract more complex features. This network consists of two linear layers with weight parameters $\boldsymbol{W}_1 \in \mathbb{R}^{D \times d_{ff}}$ and $\boldsymbol{W}_2 \in \mathbb{R}^{d_{ff} \times D}$, and bias are $b_1$ and $b_2$. The dimension of the hidden layer is $d_{ff}$. The output $\boldsymbol{Z}_{l,c}$ of the feedforward network is calculated as follows:

$$\boldsymbol{Z}_{l,c} = \phi_{GELU}(\widetilde{\boldsymbol{O}}_{l,c}\boldsymbol{W}_1 + b_1)\boldsymbol{W}_2 + b_2 \tag{16}$$

where $\phi_{GELU}(\cdot)$ is the GELU activation function. The output data $\widetilde{\boldsymbol{O}}_{l,c}$ is combined with the output data $\boldsymbol{Z}_{l,c}$ through a residual connection to address vanishing gradients and then passed through normalization layer to produce $\widetilde{\boldsymbol{Z}}_{l,c}$ for further processing in the network. The output data $\widetilde{\boldsymbol{Z}}_{l,c}$ is computed as follows:

$$\widetilde{\boldsymbol{Z}}_{l,c} = \varphi(\boldsymbol{Z}_{l,c} + \widetilde{\boldsymbol{O}}_{l,c}) \tag{17}$$

In this way, the single-band satellite image time series feature representations $\widetilde{\boldsymbol{Z}}_{l,c}$ are concatenated along the band dimension $C$ to produce the output feature representations for

all bands $Z_l = (\widetilde{Z}_{l,1}, \widetilde{Z}_{l,2}, ... \widetilde{Z}_{l,C})$.

## C. Gated Channel Attention

In crop classification tasks, neglecting the relationships between bands can lead to the loss of semantic information. Therefore, a gated channel attention mechanism is integrated into the Transformer framework to effectively capture these inter-band relationships by dynamically adjusting their weights. By assigning varying levels of importance to different bands, the channel attention mechanism enhances the significance of critical band information, thereby improving crop classification performance.

For the output data $Z_l = [Z_{l,1}, Z_{l,2}, ... Z_{l,N_l}]$, each the $n$-th $(n = 1, ..., N_l)$ patch sequence is denoted by $Z_{l,n} \in \mathbb{R}^{D \times C}$. Initially, a linear transformation is performed on $Z_{l,n}$. The weight $W_A \in \mathbb{R}^{D \times C}$ for the spectral band dimension is calculated as follows:

$$W_A = \phi_{softmax}(Z_{l,n} W_C + b_C) \tag{18}$$

where the weight parameters of the linear transformation are $W_C \in \mathbb{R}^{C \times C}$ and the bias $b_C$. The weighted output feature $\widetilde{Z}_{l,n}$ is obtained by dynamically weighting the input data $Z_{l,n}$, as follows:

$$\widetilde{Z}_{l,n} = Z_{l,n} \odot W_A \tag{19}$$

where the symbol $\odot$ denotes the Hadamard product, representing the element-wise multiplication of two matrices.

The feature representation $\widetilde{Z}_{l,n} \in \mathbb{R}^{D \times C}$ is transformed by the $1 \times 1$ convolutional layer to obtain the output $Y_{l,n}$, enhancing the capability of capturing non-linear patterns.

$$Y_{l,n} = \phi_{GELU}(\widetilde{Z}_{l,n} * W_{conv1} + b_{conv1}) * W_{conv2} + b_{conv2} \tag{20}$$

where $W_{conv1} \in \mathbb{R}^{2C \times C \times k}$ and $W_{conv2} \in \mathbb{R}^{C \times 2C \times k}$ are the weight parameters of the convolutional layers, $b_{conv1}$ and $b_{conv2}$ are bias, the symbol $*$ denotes the convolution operation, and $k$ is the convolution kernel size. The feature representations of all patch sequences are concatenated to obtain the output $Y_l = (Y_{l,1}, Y_{l,2}, ... Y_{l,N_l})$.

The input $Z_l$ and the output $Y_l$ of the convolution layer are combined through a residual connection and then passed through the normalization layer to obtain the output feature representation $\widetilde{Y}_l$. The calculation of $\widetilde{Y}_l$ is as follows:

$$\widetilde{Y}_l = \varphi(Y_l + Z_l) \tag{21}$$

where $\varphi(\cdot)$ represents the LayerNorm normalization layer. The enhanced Transformer is different from the original Transformer network described in ([15]) by incorporating the gated channel attention mechanism. In this way, the multi-scale feature can be extracted by the PatchSITS model.

## III. MULTI-SCALE WEIGHTED FEATURE FUSION CLASSIFICATION

### A. Multi-Scale Feature Fusion

The multi-scale feature fusion network is designed to provide a thorough understanding of SITS data by capturing the significance of features across different scales. This fusion network dynamically adjusts the weights of features on different scales.

For the input data $X \in \mathbb{R}^{T \times C}$, multi-scale patch feature extraction produces the multi-scale feature representations $\{\widetilde{Y}_1, \widetilde{Y}_2 ..., \widetilde{Y}_L\}$, where the patch length set is denoted by $\{P_1, P_2 ..., P_L\}$. The feature representation with the single scale $\widetilde{Y}_l \in \mathbb{R}^{N_l \times D \times C}$ is flattened to obtain $\widetilde{Y}_{l,flatten} \in \mathbb{R}^{1 \times N_l \cdot D \cdot C}$. In order to simplify processing and reduce complexity, $\widetilde{Y}_{l,flatten}$ is mapped to $D$ feature dimensions through a linear transformation. The output $\widehat{Y}_l \in \mathbb{R}^{1 \times D}$ is calculated by the following equation:

$$\widehat{Y}_l = \widetilde{Y}_{l,flatten} W_l + b_l \tag{22}$$

where the weight parameter of the linear transformation is $W_l \in \mathbb{R}^{N_l \cdot D \cdot C \times D}$. The multi-scale feature $\{\widehat{Y}_1, \widehat{Y}_2 ..., \widehat{Y}_L\}$ are concatenated to obtain the combined feature representation $\widehat{Y} = (\widehat{Y}_1, \widehat{Y}_2, ..., \widehat{Y}_L) \in \mathbb{R}^{L \times D}$. Then, the average pooling is applied along the scale dimension $L$ to obtain the feature representation $\widehat{Y}_{mean} \in \mathbb{R}^{1 \times D}$, as shown in Fig. 2:

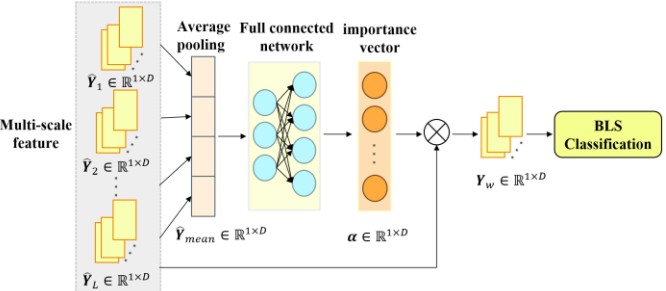

Fig. 2. Architecture of the multi-scale feature fusion network.

This mean feature representation $\widehat{Y}_{mean}$ is processed by a fully connected network to learn the importance weight at different scales. The importance weights $\boldsymbol{\alpha} \in \mathbb{R}^{1 \times L}$ is calculated by the following equation:

$$\boldsymbol{\alpha} = \phi_{sigmoid}(\phi_{GELU}(\widehat{Y}_{mean} W_3 + b_3) W_4 + b_4) \tag{23}$$

where $W_3 \in \mathbb{R}^{D \times D}$ and $W_4 \in \mathbb{R}^{D \times L}$ are the fully connected network weight parameters, with $b_3$ and $b_4$ are bias. A weighted aggregation layer is then used to obtain the multi-scale feature fusion representation $Y_w \in \mathbb{R}^{1 \times D}$:

$$Y_w = \phi_{ReLU}\left(\sum_{l=1}^{L} \alpha[l] \times \widehat{Y}_l\right) \tag{24}$$

where $\phi_{ReLU}(\cdot)$ is the ReLU activation function. The multi-scale feature fusion representation $Y_w$ is passed through a linear layer $Y_K = Y_w W_5 + b_5$ to obtain a predicted category vector $Y_K \in \mathbb{R}^{1 \times K}$, where $W_5 \in \mathbb{R}^{D \times K}$ are the linear layer weight parameters and $K$ is the number of crop classification categories.

To obtain the classification result, the predicted category vector $Y_K$ is converted into a class probability distribution $P_K = [p_1, p_2, \cdots, p_K] \in \mathbb{R}^{1 \times K}$ through the softmax function. By analyzing this class probability $P_K$ distribution, the preferences for crop classification can be identified. The prediction equation is as follows:

$$\boldsymbol{P}_K = \phi_{softmax}(\boldsymbol{Y}_K) \tag{25}$$

Thus, the predicted class probability distribution $\boldsymbol{P}_K \in \mathbb{R}^{1 \times K}$ for the satellite image time series sample $\boldsymbol{X}$ is obtained.

### B. Broad Learning System for Classification Based on Feature Fusion

Given a SITS sample set $\boldsymbol{X} = \{\boldsymbol{X}_1, \boldsymbol{X}_2, ..., \boldsymbol{X}_M\}$ with $M$ samples, the predicted probability $\boldsymbol{P}_{i,K} = [p_{i,1}, p_{i,2}, \cdots, p_{i,K}]$ for the $i$-th sample $\boldsymbol{X}_i$ is obtained through the category probability prediction using equation (25). Here, $p_{i,k}$ is the probability that the SITS sample $\boldsymbol{X}_i$ is classified into the $k$-th category. The cross-entropy loss function ([16]) for the true category distribution $q_{i,k}$ is defined as follows:

$$\mathcal{L} = L_{CE} = -\sum_{i=1}^{M} \sum_{k=1}^{K} q_{i,k} \log(p_{i,k}) \tag{26}$$

The backpropagation algorithm is employed to minimize the $\mathcal{L}$. Gradients are computed to update the model parameters $\boldsymbol{W}$, which include the weights and biases of the patch embedding layer, the enhanced Transformer layer, and multi-scale feature fusion layer. The most optimal $\boldsymbol{W}^*$ are optimized by minimizing the cross-entropy loss $\mathcal{L}$:

$$\boldsymbol{W}^* = \arg\min_{\boldsymbol{W}} \mathcal{L}(\boldsymbol{W}) \tag{27}$$

The optimization problem (27) is addressed using the AdamW algorithm ([17]), with parameter updates defined by the following equation:

$$\boldsymbol{W}_\tau = \boldsymbol{W}_{\tau-1} - \eta \left(\boldsymbol{m}_\tau / \sqrt{\boldsymbol{v}_\tau} + \varepsilon\right) - \lambda \boldsymbol{W}_{\tau-1} \tag{28}$$

where $\eta$ represents the learning rate, $\varepsilon$ represents a minimal value employed to avoid excessively large results, and $\lambda$ signifies the weight decay coefficient. The terms $\boldsymbol{m}_\tau$ and $\boldsymbol{v}_\tau$ correspond to the first-order and second-order moment estimates of the gradient $\boldsymbol{\mu}_\tau$, calculated as follows:

$$\boldsymbol{m}_\tau = (\beta_m \boldsymbol{m}_{\tau-1} + (1-\beta_m)\boldsymbol{\mu}_\tau)/(1-\beta_m{}^\tau) \tag{29}$$

$$\boldsymbol{v}_\tau = (\beta_v \boldsymbol{m}_{\tau-1} + (1-\beta_v)\boldsymbol{\mu}_\tau{}^2)/(1-\beta_v{}^\tau) \tag{30}$$

where $\beta_m$ and $\beta_v$ are the respective weight parameters. The first-order moment $\boldsymbol{m}_{\tau-1}$ and the second-order moment $\boldsymbol{v}_{\tau-1}$ are computed during the $(\tau-1)$-th iteration. The gradient $\boldsymbol{\mu}_\tau$ is defined as $\boldsymbol{\mu}_\tau = \nabla_{\boldsymbol{W}} \mathcal{L}(\boldsymbol{W}_{\tau-1})$. Finally, the optimal parameter $\boldsymbol{W}^*$ of the model is determined.

Further, BLS is selected as the classifier, as illustrated in Fig. 3. The features of the SITS sample set $\boldsymbol{X} = \{\boldsymbol{X}_1, \boldsymbol{X}_2, ..., \boldsymbol{X}_M\}$ with $M$ samples are represented by $\boldsymbol{Y}_w = (\boldsymbol{Y}_{1,w}, \boldsymbol{Y}_{2,w}, ..., \boldsymbol{Y}_{M,w}) \in \mathbb{R}^{M \times D}$.

First, the mapped features $\boldsymbol{F}_i \in \mathbb{R}^{M \times q_i}$ can be obtained by $\boldsymbol{F}_i = \phi(\boldsymbol{Y}_w \boldsymbol{W}_{F,i} + \boldsymbol{\beta}_{F,i})$ for $i = 1, ..., M_Q$, where $\boldsymbol{W}_{F,i}$ represents the weight parameters, $\phi(\cdot)$ is the nonlinear activation function, and $\boldsymbol{\beta}_{F,i}$ denotes the bias. After executing $Q$ feature mapping operations, the total feature nodes $\boldsymbol{F} = (\boldsymbol{F}_1, \boldsymbol{F}_2, ..., \boldsymbol{F}_{M_Q}) \in \mathbb{R}^{M \times Q}$ is achieved through the combination of all the mapped features.

The enhancement features $\boldsymbol{E}_j \in \mathbb{R}^{M \times r_j}$ are computed by $\boldsymbol{E}_j = \phi(\boldsymbol{F}\boldsymbol{W}_{E,j} + \boldsymbol{\beta}_{E,j})$ for $j = 1, ..., M_R$, where $\boldsymbol{W}_{E,j}$ is the weight parameters and $\boldsymbol{\beta}_{E,j}$ is the bias. After performing $R$

enhancement operations, the total enhancement nodes $\boldsymbol{E} = (\boldsymbol{E}_1, \boldsymbol{E}_2, ..., \boldsymbol{E}_{M_R}) \in \mathbb{R}^{M \times R}$ can be obtained by concatenating all enhancement features. Subsequently, the extended matrix $\boldsymbol{V} = (\boldsymbol{F}, \boldsymbol{E}) \in \mathbb{R}^{M \times (Q+R)}$ is formed by concatenating $\boldsymbol{F}$ and $\boldsymbol{E}$. The output data $\widetilde{\boldsymbol{Y}}_w$ is defined as follows:

$$\widetilde{\boldsymbol{Y}}_w = \boldsymbol{V}\boldsymbol{W}_v \tag{31}$$

where $\boldsymbol{W}_v = (\alpha \boldsymbol{I} + \boldsymbol{V}\boldsymbol{V}^T)^{-1}\boldsymbol{V}^T \boldsymbol{P}$ with $\alpha$ as the regularization parameter and $\boldsymbol{P}$ as the label matrix.

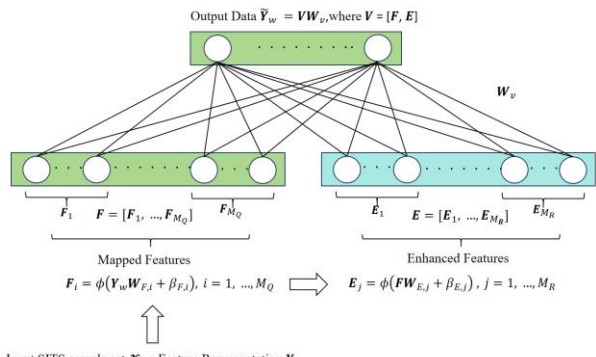

Fig. 3. Architecture of the broad learning system.

Therefore, the crop classification outputs for the SITS set $\boldsymbol{X} = \{\boldsymbol{X}_1, \boldsymbol{X}_2, ..., \boldsymbol{X}_M\}$ are derived using BLS.

## IV. EXPERIMENTS

### A. Dataset Introduction

The BreizhCrops dataset ([9]) is sourced from Sentinel-2 L1C level remote sensing time series data of the Brittany region in northwestern France from 2017. The area is split into four subareas (FRH01, FRH02, FRH03, and FRH04), encompassing a total of 768,175 parcels. Each parcel is averaged and combined into a feature vector, as illustrated by the example from the BreizhCrops dataset in Fig. 4. The feature vector contains 13 spectral bands and 45 time-steps. For model selection, FRH01 and FRH02 are utilized for training, and FRH03 serves as the validation set. For model evaluation, both FRH01 and FRH02, along with FRH03, are employed for training, leaving FRH04 as the test set.

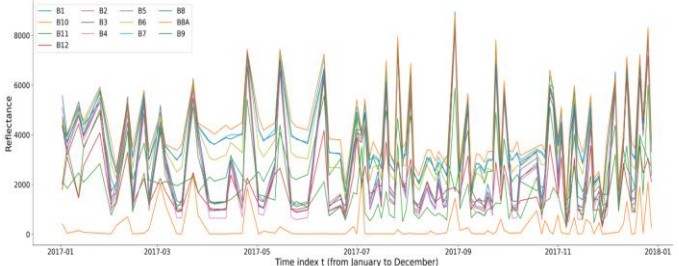

Fig. 4. Example from the BreizhCrops dataset.

This dataset mainly contains 9 crop categories: wheat, barley, corn, rapeseed, sunflower, orchard, nut, temporary meadow, and permanent meadow. The precise number of crops is presented in the subsequent table.

TABLE I.        NUMBER OF CROPS IN THE BREIZHCROPS DATASET

| ID | CropType | FRH01 | FRH02 | FRH03 | FRH04 |
|----|----------|-------|-------|-------|-------|
| 1 | barely | 13046 | 10733 | 7148 | 5978 |
| 2 | wheat | 30368 | 15005 | 27189 | 16993 |
| 3 | rapeseed | 5593 | 2346 | 3557 | 3236 |
| 4 | corn | 43990 | 36593 | 41992 | 31333 |
| 5 | sunflower | 1 | 6 | 10 | 2 |
| 6 | orchards | 944 | 350 | 1223 | 553 |
| 7 | nuts | 10 | 18 | 10 | 11 |
| 8 | permanent meadows | 32650 | 36512 | 32534 | 26117 |
| 9 | temporary meadows | 52011 | 39082 | 52728 | 38391 |
| | total | 178613 | 140645 | 166391 | 122614 |

## B. Comparison Method

The presented PatchSITS model is compared with the following six models to evaluate its classification accuracy: TempCNN, StarRNN, Bi-LSTM, Transformer, PatchTST and CA-TCN.

The hyperparameters of the TempCNN, StarRNN, Bi-LSTM, and Transformer models are consistent with those in [9]. The TempCNN model utilizes a kernel size of 7 in its convolutional layers and comprises 128 hidden units. Its architecture consists of three convolutional layers. The StarRNN model comprises of three layers with a hidden state dimension of 128. The Bi-LSTM model is composed of four stacked bidirectional LSTM layers, each with 128 hidden units. The Transformer model employs 3 stacked modules which contains 8 multi-head attention mechanisms. The PatchTST model uses a 3-layer Transformer network, and each layer consisting of 16 multi-head attention modules. The PatchTST model selects the best classification result from patch lengths of {4,8,16,32}. The CA-TCN model is constituted by four layers, each featuring 64 hidden units. Additionally, the CA-TCN model incorporates channel attention with a reduction factor of 4. The primary parameters utilized in the experiments for the PatchSITS model are presented in Table II.

TABLE II.        MAIN PARAMETER CONFIGURATIONS OF THE EXPERIMENTS

| Parameter | Value | Description |
|-----------|-------|-------------|
| $E$ | 100 | The total training epochs |
| $B$ | 64 | The size of each batch |
| $N$ | 4 | Number of layers in Transformer module |
| $\eta$ | 0.001 | The learning rate of the optimizer |
| $P$ | {3,4,6} | The multi-scale patch set |
| $d$ | 0.1 | The dropout rate parameter |
| $h$ | 16 | The multi-head number |
| $D$ | 128 | The dimensionality of the hidden layers |

All deep learning models are implemented using the PyTorch and optimized with the AdamW method. The training procedure is executed over 100 epochs. The efficacy of the classification results is assessed through the use of three key metrics: overall accuracy (OA), the Kappa coefficient, and the average F1 score. The Kappa coefficient assesses the consistency between the classification results and the expected outcomes derived from random chance. The mean F1 score is computed by averaging the F1 scores. The results of the

TABLE III.        CLASSIFICATION RESULTS ON THE BREIZHCROPS DATASET

| Class | Temp CNN | Star RNN | BiLS TM | Transf ormer | Patch TST | CA-TCN ([5]) | Patch SITS |
|-------|----------|----------|---------|--------------|-----------|--------------|------------|
| 1 | 93.71 | 92.67 | 91.50 | 92.65 | 86.99 | 94.01 | **94.81** |
| 2 | 96.86 | 97.77 | 97.92 | 96.82 | 94.07 | 97.59 | **97.92** |
| 3 | 96.66 | 95.77 | 97.23 | 96.91 | 95.02 | **97.58** | 97.03 |
| 4 | 97.67 | **97.99** | 97.59 | 97.45 | 96.47 | 97.78 | 97.62 |
| 5 | 0.00 | 0.00 | 0.00 | 0.00 | 0.00 | 0.00 | 0.00 |
| 6 | 4.33 | 0.00 | 7.78 | 7.41 | 0.00 | **16.27** | 12.12 |
| 7 | 0.00 | 0.00 | 0.00 | 0.00 | 0.00 | 0.00 | 0.00 |
| 8 | 49.37 | 53.51 | 57.21 | **61.46** | 46.43 | 57.92 | 58.04 |
| 9 | **78.08** | 72.82 | 72.62 | 70.93 | 75.64 | 73.80 | 74.68 |
| OA | 80.48 | 79.87 | 80.50 | 80.73 | 78.01 | 81.20 | **81.36** |
| Kappa | 74.46 | 68.93 | 74.59 | 74.91 | 71.29 | 75.52 | **75.69** |
| mF1 | 57.55 | 50.34 | 58.42 | 58.84 | 54.10 | 57.76 | **59.80** |

The classification results for all models on the BreizhCrops dataset are presented in Tab. III. Compared to the baseline models, the PatchSITS model achieves the best overall accuracy, Kappa score, and mean F1 score. This experimental result indicates that the PatchSITS model can effectively capture the multi-scale time-dependent features and improve the crop classification accuracy.

## C. Hyperparameter experiment

In the PatchSITS model, a series of hyperparameter experiments are conducted to explore the parameters that can influence the classification accuracy, including the number of layers and attention heads in the Transformer model. The model layers are systematically varied among the values {1,2,3,4}, while the number of attention heads is evaluated at {4,8,16,32}. The learning rate is set to 0.001 and the dropout rate is set to 0.1. The results of the hyperparameter experiment are presented in Fig. 5.

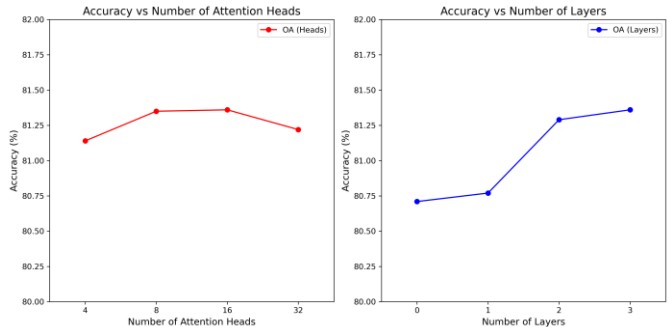

Fig. 5.   Results of the hyperparameter experiment.

The experimental results show that the classification accuracy is best with 4 layers and 16 attention heads. As the number of attention heads increases, the PatchSITS model can better capture the key features and the complex relationships in SITS data. Meanwhile, increasing the number of Transformer layers can enhance both the representational capacity and the feature extraction capacity of the model, thereby improving the crop classification accuracy.

## D. Ablation Experiments

In addition, an ablation study is conducted to evaluate the

impact of gated channel attention (GCA) and multi-scale feature fusion (MSF) on the performance of the PatchSITS model. Three types of variant models are designed: removal of gated channel attention, removal of multi-scale feature fusion, and removal of both. Simulation experiments are performed on the BreizhCrops dataset. In the Tab. IV, √ represents the use of the corresponding module, and × represents the removal of the corresponding module.

TABLE IV.    ABLATION EXPERIMENT RESULTS

| Module | | Performance metric | | |
|---|---|---|---|---|
| GCA | MSF | OA | Kappa | mF1 |
| × | × | 80.10 | 73.97 | 57.74 |
| √ | × | 80.39 | 74.42 | 58.23 |
| × | √ | 80.46 | 74.50 | 58.25 |
| √ | √ | **81.36** | **75.69** | **59.80** |

According to the results presented in Tab. IV, the PatchSITS model achieves the highest scores in overall accuracy (OA), kappa coefficient, and average F1 score when both GCA and MSF modules are employed. This ablation study demonstrates that the integration of both the GCA and MSF modules can enhance the classification accuracy by effectively capturing and fusing critical features within the dataset.

## V.  CONCLUSIONS

This paper proposes a multi-scale patch Transformer model for SITS classification. By extracting multi-scale features, both local and global temporal information are effectively utilized to improve classification performance. Future research will focus on more efficient multi-scale segmentation methods for SITS classification.

## ACKNOWLEDGMENT

This work was supported by the National Key Research and Development Program of China (2018AAA0100300) and the National Natural Science Foundation (NNSF) of China (12071056).

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
