# OpenReview forum: "Multi-Scale Patch Transformer Network for Satellite  Image Time Series Classification"
_IEEE.org/ICIST/2024/Conference — IEEE ICIST 2024 Conference Submission_

### Official Review · Reviewer_9dJB · 2024-08-21
**Accept**

**Rating:** 7
**Confidence:** 4

**Review:**

This paper introduces a multi-scale patch transformer model, PatchSITS, for crop classification using satellite image time series. It segments images into sequences of patches and employs a transformer to extract multi-scale features, enhanced by a gated channel attention mechanism for inter-band relationships. Numerical simulations demonstrate the model's effective classification performance on the BreizhCrops dataset.

Are there any limitations or assumptions in proposed PatchSITS model that might affect its applicability to enhance crop classification? What futher research is needed to build on your findings?

How can the optimal fusion weights for multi-scale features within the PatchSITS framework be determined, and what impact do these weights have on the classification performance?

When processing remote sensing data of varying resolutions, how can the PatchSITS model ensure consistent and stable performance across different scales?

What are the advantages and limitations of the enhanced Transformer module used in the model compared to other Transformer models?

---

### Official Review · Reviewer_Jb27 · 2024-08-24
**The paper is written clearly, exceptionally excellent**

**Rating:** 8
**Confidence:** 3

**Review:**

This paper excels in terms of quality, clarity, originality, and significance, but I would still like to offer some suggestions.
1. In model evaluation part, Is there a better way?
2. Elucidate how the work in this paper relates to your subsequent work.

---

### Official Review · Reviewer_g2Wn · 2024-08-25
**add sufficient background**

**Rating:** 8
**Confidence:** 4

**Review:**

In the manuscript titled "Multi-Scale Patch Transformer Network for Satellite Image Time Series Classification",Jifeng Suo et al.performed the PatchSITS model, which uses multi-scale patch sequences, an enhanced transformer, gated channel attention, and a multi-scale weighted fusion strategy to achieve effective crop classification from satellite image time series. This study contains some interesting findings and are valuable for the understanding of to improving classification performance.However, lack of sufficient domestic and foreign literature is the major flaw of the study .It is recommended to add sufficient background and literature comparison.

---

### Official Review · Reviewer_w24W · 2024-08-25
**In this paper, a multi-scale patch transformer model (PatchSITS) is proposed, and enhanced transformers are designed in combination with k-means clustering to extract local and global multi-scale features. In order to capture the relationship between frequency bands and enhance the information of key frequency bands, the gated channel attention mechanism is used to obtain the dynamic weights between frequency bands. Finally, the multi-scale weighted fusion strategy is used to fuse these multi-scale features, and the BLS is considered to classify the sit. The model achieves good classification results on the BreizhCrops dataset. However。 1. Highlight the originality of the work. 2. Whether the future work is explained.**

**Rating:** 7
**Confidence:** 4

**Review:**

The research content of this paper is that the use of satellite image time series (SITS) for crop classification has become a research hotspot as high-quality Earth observation data become more and more readily available. The effective use of time information has become a key challenge. The solution is to propose a multi-scale SMD transformer model (PatchSITS). Firstly, k-means clustering was used to segment the SIT samples into patch sequences with different patch lengths. Subsequently, enhanced transformers were designed to extract local and global multi-scale features. In order to capture the relationship between frequency bands and enhance the information of key frequency bands, the gated channel attention mechanism is used to obtain the dynamic weights between frequency bands. Finally, the multi-scale weighted fusion strategy is used to fuse these multi-scale features, and the BLS is considered to classify the sit. Numerical simulation results show that the model achieves good classification effect on the BreizhCrops dataset.
However。 1. Highlight the originality of the work.
2. Whether the future work is explained.

---

### Decision · Program_Chairs · 2024-09-08

Accept (Oral)